# Agitation near the end of life with dementia: An ethnographic study of care

**Elizabeth L. Sampson**[1]*, **Aisling Stringer**[2], **Francesca La Frenais**[2], **Shanlee Higgins**[3], **Mary-Jo Doyle**[3], **Anne Laybourne**[2], **Gill Livingston**[2], **Gerard Leavey**[4]

1 Marie Curie Palliative Care Research Unit, Division of Psychiatry, University College London, England, London, United Kingdom, 2 Department of Old Age Psychiatry, Division of Psychiatry, University College London, London, England, United Kingdom, 3 Camden and Islington NHS Foundation Trust, London, England, United Kingdom, 4 Bamford Centre for Mental Health and Wellbeing, School of Psychology, Ulster University, Coleraine, Northern Ireland, United Kingdom

* e.sampson@ucl.ac.uk

## Abstract

### Background and objectives

Agitation is common in people living with dementia especially at the end of life. We examined how staff interpreted agitation behavior in people with dementia nearing end of life, how this may influence their responses and its impact on the quality of care.

### Research design

Ethnographic study. Structured and semi-structured non-participant observations (referred to subsequently in this paper as "structured observations") of people living with dementia nearing the end of life in hospital and care homes (south-east England) and in-depth interviews with staff, conducted August 2015-March 2017.

### Methods

Three data sources: 1) detailed field notes, 2) observations using a structured tool and checklist for behaviors classed as agitation and staff and institutional responses, 3) staff semi-structured qualitative interviews. We calculated the time participants were agitated and described staff responses. Data sources were analyzed separately, developed continuously and relationally during the study and synthesized where appropriate.

### Results

We identified two main 'ideal types' of staff explanatory models for agitation: In the first, staff attribute agitated behaviors to the person's "moral judgement", making them prone to rejecting or punitive responses. In the second staff adopt a more "needs-based" approach in which agitation behaviors are regarded as meaningful and managed with proactive and investigative approaches. These different approaches appear to have significant consequences for the timing, frequency and quality of staff response. While these models may overlap they tend to reflect distinct organizational resources and values.

**Data Availability Statement:** All relevant data are available within the paper and in the Mendeley Data at https://data.mendeley.com/datasets/jcz7wcs8vd/2.

**Funding:** This study was funded by a grant from the UK Economic and Social Research Council and the National Institute of Health Research Grant number NIHR/ESRC ES/L001780/1. ELS and GL are supported by the UCLH NIHR Biomedical Research Centre. ELS is supported by Marie Curie core grant funding [MCCC-FCO-16-U]. The funders had no role in study design, data collection and analysis, decision to publish, or preparation of the manuscript.

**Competing interests:** The authors have declared that no competing interests exist.

## Conclusions

Care worker knowledge about agitation is not enough, and staff need organizational support to care better for people living with dementia towards end of life. Positional theory may help to explain much of the cultural-structural context that produces staff disengagement from people with dementia, offering insights on how agitation behavior is reframed by some staff as dangerous. Such behavior may be associated with low-resource institutions with minimal staff training where the personhood of staff may be neglected.

## Introduction

Agitation in dementia, as it is currently understood, affects over 40% of care home residents [1] and 75% of older medical hospital in-patients with dementia [2]. Agitation is broadly defined as restlessness, pacing, shouting, and verbal or physical aggression; it is complex and multifactorial with a range of biological, psychological and social causes. It may be a direct result of neurodegeneration, affecting brain circuits that control behavior, and also an expression of unmet needs (e.g. pain or thirst, lack of communication or comfort) [3], indicating emotional distress [4]. Sometimes deemed as aggression or challenging behavior, agitation can be difficult, harmful and exhausting for patients and carers [5]. Thus, the impact on nursing staff and services include burnout, sick leave, turnover, and increased economic costs. Agitation is a common and distressing issue which affects between 20–54% of people with severe dementia [6, 7]. The mean Cohen Mansfield Agitation Inventory CMAI) score for people with severe dementia living in care homes is 48.3 (standard deviation 19.7) [1] indicating a clinically significant level of distress. Because most people with dementia die in care homes or acute hospitals [8] it is crucial we explore how staff in these settings understand and respond to agitation near the end of life.

Some evidence suggests that hospital and care home staff are ambivalent in their interpretation of agitation, shifting between seeing agitation as a response to 'unmet needs' or thinking of agitation as a 'meaningless' epiphenomena which is just "part of dementia". In the latter scenario, staff may struggle with the paradox of considering agency and personhood in a person with dementia. For example, that the person with dementia is responsible for their own actions, whilst at the same time not blaming the person with dementia for 'bad' or 'challenging behavior' [9]. However, it can be argued that personal and cultural responses to agitation and quality of care, more generally, are also determined by the social and organizational context of care, institutional support, the environment, staff morale, training, and work pressure, among other things [10]. Staff's own personhood may not be acknowledged and this can reduce the complexities of care work to a series of tasks, challenge care workers' self-worth and self-efficacy, and impede their efforts to deliver person-centered care [11]. The relationships between these factors and agitation in people with dementia nearing end of life have not been examined.

## Aims

To explore how agitation in people with dementia nearing the end of life is understood and managed in different care settings.

## Methods

### Ethics

The study was approved by London South Ethics Committee 15/LO/0222. The ward or care home manager signed a collective consent form agreeing researchers could conduct

observations and, during these, observe any staff member present on the ward. Information sheets were given to staff members.

Our participants had severe dementia or had moderate dementia and were physically unwell, thus they lacked capacity to give consent to participate in the research project. Therefore our consent process for the structured observations complied with English and Welsh capacity legislation (Mental Capacity Act 2005, Sections 30–34). This requires us to speak with a "personal consultee"- a family member or other key close person for the person with dementia. They are required to advise on what the person's wishes would have been on participating in research when they had capacity and sign a declaration regarding this. If no family member was available we used a professional "nominated" consultee. The researcher checked intermittently with those present that they had no objections to their presence, particularly during personal care. Hospital and care home staff gave written informed consent to participate in the qualitative interviews.

The study and potential ethical issues arising from this was discussed with hospital and care home managers who informed staff of our presence and that they could withdraw from interviews or refuse to be observed at any time. There was a written protocol in place, approved by the ethics committee, should the research team have ethical concerns during the conduct of the study. If researchers had any concerns regarding the treatment of participants in the hospitals or care homes they were able to immediately contact and discuss this with a senior clinical researcher in the team. If appropriate, they approached the participant to seek their consent for disclosure. If the participant lacked capacity to give this consent the research assistant approached their personal or nominated consultee. The information sheet specified that "we respect confidentiality but cannot keep it a secret if anyone is being seriously harmed or is at high risk of serious harm". If there was reason to believe that harm was occurring or there was a high risk it was likely to occur, we reported this to the hospital ward or care home manager even if consent was refused. These procedures complied with standard English safeguarding laws and practices for vulnerable adults.

## Design

We chose an ethnographic approach as this is appropriate to explore and identify social interactions, activities, and beliefs that exist within social groups and communities. It has been widely used to explore the dynamics of teams and organizations. Thus, ethnography can provide rich insights into what people do, how they do it and the context in which they live and work. In keeping with an ethnographic approach we conducted structured observations of people with dementia within hospitals and care homes and in-depth interviews with staff. The objectives of the observations were to:

1. Explore staff appraisal of the agitation, their causal attributions and coping styles.

2. Examine responses to agitation behavior (time spent with person with dementia and type of response).

3. Explore how staff perceptions and behaviors influence response to agitation.

4. Quantify the frequency and types of agitation observed within care home and hospital settings.

The objective of the staff interviews was to explore how staff understood and interpreted agitation.

## Study settings

We purposively selected two English hospitals in socio-demographically and ethnically varied areas, and two care homes to provide diversity in terms of size, ownership status, location and residents. Data were collected between August 2015 and March 2017.

## Participants

**People with dementia.**  We aimed to observe 20 people with dementia (diagnosis in clinical notes or scored as cases of dementia in the validated Noticeable Problems Checklist) [12]. This is a six item questionnaire (with items scored 0 or 1) designed for use by care staff with little specific training, to identify people with likely cognitive impairment. Higher scores suggest increased cognitive problems. We selected those likely to nearing the end of life: Severe dementia (Fast stage 6c or higher: Urinary or faecal incontinence, through to limited ability to speak, ambulate, sit up or smile) [13], or a recorded diagnosis of moderate dementia and frailty with an inter-current medical illness, for example pneumonia, or other acute medical event [14]. We excluded participants who indicated verbally or non-verbally that they did not wish to participate, those with delirium without a previous diagnosis of dementia, and those where there were clinical or social concerns that precluded them being approached.

**Care home and acute hospital staff.**  We aimed to identify 20 staff members in these settings, purposively selecting a range of occupations; healthcare assistants, nurses, doctors, allied healthcare professionals and domestic staff and a range of ages, sex, ethnicities.

## Data collection

**Researchers.**  Four female researchers, aged 25–38 years; two of whom were psychiatric nurses and two psychology graduates, all experienced in dementia research and previously worked in dementia care settings, conducted observations. Researchers reflected on the observations and the research challenges (such as non-intervention for distressing events, potential ethical and legal concerns, and clinical concerns) with fellow team members and senior investigators.

**Fieldwork observations of the settings.**  Researchers, assigned to one or several sites, spent up to two weeks building relationships with staff. This enabled the researcher to reassure staff about the nature of the study and for their presence to become unexceptional. They made detailed field note observations of settings such as physical layout, décor, and general mood.

**Staff interviews.**  Digitally recorded interviews, in a private room, lasted a maximum of one hour. To explore how staff understood and interpreted agitation we adapted Kleinman's explanatory models interview [15]: A way of understanding a person's ideas about the nature of the problem, its cause, severity and treatment (for topic guide see appendix 1). Interviews were professionally transcribed and anonymised, ensuring that all potentially identifying details were removed.

**Structured observations.**  Observations were conducted to capture a range of activity during the day, for example on waking, during morning personal care, during the morning rest period and until lunchtime. If participants slept all day, observations were carried out in the evenings and into the night. This ensured the collection of rich data on interactions and allowed observations where participants' sleep-wake cycle was disturbed. Researchers made notes at the start of each structured observation, about staffing levels, number of patients/residents present, ambience (noise/lights), and level of activity around the participant. During observations, field notes were made describing all activity and any engagement from staff, family members, or other patients/residents. Field notes were taken detailing anything happening in the environment that may affect the participant.

We developed an instrument for the observations to describe participant behavior at two-minute intervals, recording (1) whether asleep, awake, or engaged in an agitation behavior; (2), if agitated, the type of agitation using a checklist of behaviors derived from the Cohen-Mansfield Agitation Inventory (CMAI) [16] (Appendix 2); (3) time and duration of events; (4) staff response (coded as tactile comfort, verbal comfort, chastising, or looking but not intervening. Some responses occurred simultaneously, for example tactile and verbal comfort and these were recorded as two separate responses. If an additional agitation type occurred when staff were already 'responding' and this prompted further response (based on freehand notes), then it was recorded as an additional incident and response. Extra behaviors were added to the case report forms if observed and after team discussion. Each participant's care was observed for one to four periods lasting up to four hours each (considering participant and/or family preferences), sufficient to cover a range of interactions but not too onerous for participants. During structured observations the researcher sat nearby but had no contact with the person. Prior to observations, staff provided contextual details about the person including current needs (diagnoses and painful conditions, medications, comfort measures including feeding and nutrition, sedation), presence of family and/or friends, and relationships between staff. Researchers concluded with a note about their subjective impression of the observation.

## Analyses

The primary focus of our analysis was agitation and a deeper understanding of institutional response including an interpretation of the phenomena, the emotions aroused, the role of personhood and coping strategies and resources used.

**Fieldwork observations of the settings.** After each observation visit, notes from the fieldwork were transcribed. These informed site descriptions and provided context and examples to support the data from the structured observations and staff interviews.

**Staff interviews.** We used a qualitative software package (NViVO) to assist in the organization and management of the transcribed interviews which were coded by individual researchers who grounded the analysis and the development of theory within the specific observed 'events'.

**Structured observations.** Data from the observations were analyzed using STATA [17]. We computed the amount of time each participant spent engaged in an agitation behavior, how staff responded to the agitation (length of time and type of response), and the amount of 'social' time (i.e. not a care task or agitation response) that staff spent with residents. Observations were discussed between researchers, both sporadically and during regular meetings, and informed analysis of the interviews.

## Data synthesis

The three sources of data; structured observational coding of behaviors, observational notes and qualitative interviews provided areas for separate analytical work but were continuously developed relationally and synthesized where appropriate. The researchers read each transcript several times (immersion) prior to rigorous and systematic coding which we approached without an *a priori* conceptual framework. Thus, the researchers analyzed transcripts and fieldwork data using well-recognized approaches that entailed open coding, indexing and memoing (brief notes on the meaning and potential significance of the individual codes) [18]. The coding frameworks that emerged from individual researchers' analyses were then examined to establish an agreed and coherent coding framework. Additionally, to assist the team in understanding various patterns and linkages we explored the data through the use of working documents; graphs, networks and diagrams. These exploratory processes allowed us to grapple with

possible linkages and theoretical models. Thus, we iteratively developed a theoretical framework. The preliminary observations of individual researchers were compiled and brought to meetings where these were discussed in relation to context, possible interpretations and alternative explanations, and their wider significance. Thus, each area of analysis assisted in the further development of theory and indicated new areas or perspectives for exploration.

## Findings

First, we report recruitment data. Second, sites are described based on researcher perspectives and non-structured field notes. Third, data from staff interviews are provided in order to illuminate particular themes and issues. Fourth, structured observation data are presented. We then describe two broad explanatory models of agitation derived from the data synthesis.

### Recruitment

Two hospitals and two care homes were recruited. Across these sites we recruited twenty participants with dementia. Twenty-two staff members were interviewed: Nursing, medical, therapy, healthcare assistants and housekeeping staff. Tables 1 and 2 describe demographics for participants and staff, respectively.

### Non-structured field notes

**Hospital A.** Hospital A was an outer-city district general hospital in a multicultural area. Data were collected in three single-sex older adult wards: One male and two female. All three had a similar layout: Five-bedded bays, several individual side-rooms, and a day-room in the middle. The nurses' station was on the corner of the L-shaped ward, with a wraparound desk and chest-high countertop. The décor was minimal and dull. Above the beds, the patients had a board with their full and preferred name, and physical needs (e.g. assistance with activities of daily living).

Wards were based in the old hospital annex and were always busy; often reliant on agency staff, tacit knowledge of the wards and continuity of care was limited. Staff described morale as low and often talked negatively to each other. On occasion, we witnessed staff arguing. There were frequently no staff in a bay and on one occasion no one came into a bay with five male

**Table 1. Characteristics of staff interviewed.**

| Site | Gender | Ethnicity | Role |
|------|--------|-----------|------|
| A | 4 Male<br>5 Female | 1 Black British<br>1 Black West Indian<br>1 British Asian<br>2 Black Caribbean<br>1 South Asian<br>3 White British | 1 Junior Doctor<br>3 Healthcare Assistants<br>1 Housekeeper<br>1 Physiotherapist<br>1 Physiotherapy Assistant<br>1 Staff nurse<br>1 Ward manager |
| B | 5 Female | 3 White British<br>2 Black African | 1 Clinical Practice Facilitator<br>1 Healthcare Assistant<br>2 Senior Nurses<br>1 Staff Nurse |
| C | 3 Male<br>2 Female | 2 Black African<br>2 White British<br>1 British Asian | 1 Healthcare Assistant<br>1 Music Therapist<br>2 Mental Health Nurse<br>1 Student Nurse |
| D | 3 Female | 2 Black British<br>1 Black African | 1 Deputy Manager<br>1 Healthcare Assistant<br>1 Housekeeper |

**Table 2. Participant demographic characteristics, number of observations and time spent agitated.**

| Site | Gender* | Age | Ethnicity* | English as first language? | Number of observations | % of time observed spent agitated |
|------|---------|-----|-----------|---------------------------|------------------------|-----------------------------------|
| A | 2 Male 4 Female | 92 | 1 Turkish-Cypriot 1 Black Caribbean 1 White Irish 2 White British 1 Other | No | 3 | 18.1 |
| A | | 86 | | No | 4 | 17.5 |
| A | | 92 | | Yes | 4 | 39.2 |
| A | | 93 | | Yes | 4 | 35.7 |
| A | | 97 | | Yes | 4 | 25.9 |
| A | | 81 | | Unknown | 2 | 33.3 |
| Summary, mean (SD) | | 90 (5.8) | | - | Total 21 | 28.3 (9.2) |
| B | 4 Female 2 Male | 90 | 3 White British 1 South Asian 2 White Irish | Yes | 1 | 7.6 |
| B | | 90 | | No | 1 | 12.9 |
| B | | 87 | | Yes | 1 | 53.3 |
| B | | 84 | | Yes | 3 | 17.2 |
| B | | 71 | | Yes | 2 | 64.7 |
| B | | 80 | | Yes | 2 | 12.5 |
| Summary, mean (SD) | | 84 (7.3) | | - | Total 10 | 28.0 (24.4) |
| C | 3 Male 2 Female | 82 | 1 Somali 1 Jamaican 1 White British 1 Black-African 1 Afro-Caribbean | No | 3 | 43.1 |
| C | | 91 | | Yes | 2 | 29.8 |
| C | | 87 | | Yes | 3 | 19.5 |
| C | | 78 | | Yes | 2 | 48.3 |
| C | | 91 | | Yes | 2 | 43.7 |
| Summary, mean (SD) | | 86 (5.7) | | - | Total 12 | 36.9 (11.9) |
| D | 2 Male 1 Female | 84 | 3 White British | Yes | 3 | 33.0 |
| D | | 85 | | Yes | 3 | 38.3 |
| D | | 89 | | Yes | 2 | 19.2 |
| Summary, mean (SD) | | 86 (2.6) | | | Total 8 | 30.2 (9.9) |

*Data in these columns are aggregated to ensure confidentiality. SD = standard deviation.

patients for two hours. Other needs were not met; for example call-bells were commonly out of reach, and water cups often empty. Agitation behavior tended to be ignored (even when staff were present), unless there was a reason for intervention such as personal care. We noted that staff knew little of patients' social histories. Some workplace routines seemed insensitive to patient needs: The nursing handover was conducted by walking around the bays, turning on lights while patients were asleep, and audibly discussing confidential information in public areas.

**Hospital B.** Hospital B was a teaching hospital in a large city. Data were collected in two mixed older adult wards. Ward 1 had 60 beds for older people. There were bays of four to five beds including a bay near the staff desk area specifically for patients at risk of falls (described by staff as the 'at risk' bay), several side rooms, and a reminiscence room. Ward 2 was smaller than ward 1, comprising two or three four-bedded bays on another floor (bays were used flexibly to meet the needs of the hospital). There were no nurses' stations in either ward but an area with desks, shared by all healthcare professionals. The wards were modern and clean with good natural lighting and had a calm atmosphere. Each patient had a television and "forget-me-not" magnet on their board (visual indicator used in UK hospitals to highlight patients with cognitive problems), most had a '5 things about me' board visible near them.

There was a dedicated dementia lead in the hospital, and patients with dementia and agitation often had a dedicated staff member ('specials') with a nursing background. Wards were

busy but staff did not seem rushed. The ward sister and nurses were more engaged in direct care and responding to agitation than in Hospital A, and exhibited much less fear when caring for agitated patients. Agitation was referred to as 'distress behaviors' and all staff we spoke to had received dementia training.

*"So our training tends to enlighten us on how to equip ourselves with the skills to make patient more comfortable, more relaxed, more trusting so that at least it will help them in their day to day activities." [Hospital B, staff nurse 402]*

**Care home C.** Care home C was a nursing home within a NHS mental health trust providing care for residents aged 55 years and above. The unit had 20 *en-suite* single bedrooms over two floors with communal dining and living space on each floor. Residents had long-standing mental health conditions and/or dementia diagnoses with complex needs, many had been living in the home for some years and were known very well by staff. In their bedrooms, the residents had their life stories displayed, topics they enjoyed talking and hearing about, and individually chosen bed linen, decorations, art work, and photographs. There was a lounge and dining area and quiet room on both floors, that had large fish tanks and facilities for playing music. There was a garden for activities and social events. Staff had detailed knowledge of the residents.

**Care home D.** Care home D was a charity-owned care home providing nursing, dementia, and respite care to over 65s. Data were collected from one nursing floor for residents with severe dementia. The home has more than 100 spacious *en-suite* bedrooms, over four floors with lounges on each floor and communal spaces on the ground floor. Bedrooms were situated on long corridors around the edge of the building. During the day most residents attended organized activities and mealtimes, and therefore most observations were based in two adjoining communal rooms (both large and open-plan).

Care home D felt calm, and the staff well-supported. There was usually at least one resident who was agitated, and staff seemed familiar with these behaviors. They did not ignore it, but also did not always respond. We observed that where these behaviors were longstanding and staff did not feel they were able to stop them, they also did not appear offended or challenged by them the way that staff did at Hospital A. Care home D staff ignored a lot of persistent agitated behaviors, but appeared to intervene sporadically, when addressed directly by the participant, or when agitation escalated. Staff in Care home D did not express a sense of fear of residents in their interviews.

### Staff interviews—Key themes

**Institutional context.** Researchers observed distinct differences in the quality and delivery of care between sites. From our in-depth staff interviews, we noted a spectrum of attitudes and beliefs towards dementia and agitation. We discerned differences among staff that were associated with specific institutional contexts.

The sites, particularly the hospitals, had very different levels of institutional support. Hospital A was understaffed with a pressured environment and minimal staff dementia-training. In this setting, there was a prevalent perception that unsuitable and unqualified staff are "dumped" on older people's wards due to the stigma of working with older people, and pooled recruitment practices.

The staff in the different settings reported considerable disparity in the quantity and quality of training provided. While some staff at Site A were compassionate, we observed during

interviews and time spent at the sites that even those who chose to work there lacked training and appeared to be very stressed and perhaps at risk of experiencing burnout.

*"Because of financial restrictions or whatever. . .if 80% of your patients are dementia patients with aggression, there's no way that's going to be enough staff. And on my ward as well, there are only two men on the whole team." [Hospital A, HCA 102]*

*"We are really suffering, we really need more staff in our care of elderly wards because we have a lot to do." [Hospital A, staff nurse 101]*

**Training.** Training can help staff members to think more about agitation as a response to unmet needs and allied healthcare professionals at Hospital A, who had received dementia training, reflected this:

*"Once upon a time, my, not opinion, but my phrase used to be, I treat everybody the same–but I don't, now; I treat everybody as an individual. . . .everybody's got a story to tell if you've got the time to listen" [Hospital A, Physiotherapy assistant 002]*

**Stigma.** Finally, feelings of lack of control and stigmatization of older people with dementia mean that staff become unmotivated and unable to provide good care. In contrast to other sites, who were generally quite enthusiastic and proud, staff at hospital A described how this work was seen as burdensome, "unfashionable", unrewarding and "undesirable" by other staff who have a low opinion of what they do. This was underpinned by feeling unsupported by the wider hospital organization in terms of recruitment practice, time and staffing levels, and emotional support.

*"People say oh, care of elderly, I can't work there, no. I challenge them, you know, I do challenge them. I say no, you should not say that because I look after the care of elderly. Most of the nurses here, oh, I don't like to work in care of elderly wards, no, it's crazy, you know."* *[Hospital A, staff nurse 101]*

## Structured observations

Researchers conducted a total of 52 individual observation periods and the total time observed was 7024 minutes (approximately 117 hours). The median number of observations per participant was 2.5 (IQR 2–3). We observed nine males and eleven females (see Table 1).

**Agitation.** The length of time participants spent agitated during the total observation period varied from 8 to 302 minutes (see Table 3). The participants spent a median of 31.4% (range 7.6% to 64.7%) of their total observation time agitated. The commonest behaviors observed were restlessness, hitting, strange noises, repetitive mannerisms (9 instances of each), verbal aggression and resistance to care (8 instances), repetitive sentences of speech (7 instances), and grabbing, and complaining & negativism (6 instances).

**Staff response to agitation.** In hospital A, all staff groups rejected the management of agitation as an integral aspect of nursing care. This translated into either a lack of staff response, as seen in observations, or (as described in interviews) perceived limited options in responding, including a reliance on hospital security. Thus hospital A was an outlier in terms of the percentage of time spent by staff responding during the agitated period-12.0% versus hospital B, 79.4%, care home C, 58.6% and care home D, 53.4%.

**Table 3. Staff response to agitation.**

| Site | Total time observed | | | | Comfort: Speech | | Comfort: Tactile | | Chastising: Speech | | Looking but not intervening | |
|---|---|---|---|---|---|---|---|---|---|---|---|---|
| | Total time observed (minutes) | % of the total observed time where participants were agitated | % of observed time where participants were agitated, and staff responded to this | % of observed time where participants were agitated, and staff did not respond | % of response | % of agitation receiving response | % of response | % of agitation receiving response | % of response | % of agitation receiving response | % of response | % of agitation receiving response |
| A | 3780 | 28.3 | 12.0 | 88.0 | 54.7 | 6.5 | 9.4 | 1.1 | 31.3 | 3.7 | 21.9 | 2.9 |
| B | 1138 | 29.0 | 79.4 | 20.6 | 38.2 | 30.3 | 4.6 | 3.6 | 0.8 | 0.6 | 0.8 | 0.4 |
| C | 1254 | 37.8 | 58.6 | 41.4 | 84.2 | 49.4 | 77.0 | 45.1 | 2.2 | 1.3 | 4.3 | 2.5 |
| D | 852 | 31.2 | 53.4 | 46.6 | 70.4 | 37.6 | 29.6 | 15.8 | 2.8 | 1.5 | 23.5 | 14.3 |

Please note row percentages do not add up to 100 as some types of responses occurred simultaneously, for example speech and tactile comfort and so these were double-coded

The modes and quality of staff responses to agitation varied between settings. For example, comforting responses were rare in hospitals, whilst in the care homes, tactile comfort was commonly used. At hospital A, managing dementia and associated agitation was not seen as part of the job, but more of a nuisance to be managed by 'specials' or security staff. Agitation behavior was often ignored. "Looking but not intervening" was a typical response at hospital A, but also observed with care home D.

Observation at hospital A (0306/04): "I want to go to the toilet" . . .20 seconds. . . "I want to go to the toilet but I can't go" Doctor by trolley in front of her bed. Junior doctor talking to (patient in adjacent bed), staff nurse with bed on other side. All ignore. "Let me go. . . what if you've got to go to toilet. . .got to go to toilet".

**Types of staff caring for people with agitation.** Staff spent 1390 minutes (19.8% of total observed time) interacting with patients. Health care assistants (HCAs) provided most time (30.1%) with agitated patients, followed by staff nurses (26.8%), and "specials" (also known as enhanced care or 1:1 health care assistants or nurses who are usually casual staff) (14.7%). The remainder comprised other individuals who interacted with patients during periods of agitation such as student nurses, ward assistants, and volunteers. Staff spent 5.8% of the total observation time of 7024 minutes in social time with participants. This differed between the sites with 2.5% social time at hospital A, 10.7% at hospital B, 10.4% at care home C and 8.9% at care home D.

## Explanatory models of agitation

Data analysis and synthesis suggested two major typologies reflecting explanatory models of agitation. Our models should be seen as 'ideal types' as developed by German sociologist Max Weber. 'Ideal' types were used by Weber as heuristic devices (rules-of-thumb) and in this context do not mean "ideal" as understood in the common-sense understanding of the word. An 'ideal type' encapsulates characteristics and essential components of the phenomena in question. Thus in our study settings, behaviors and attitudes were not completely demarcated but certain ideal types were distinctly associated with some care units, rather than others. In our interviews, rather than a consistent articulation of the two individual models, we noted a range of overlapping attitudes with features of both models sometimes expressed by the same participant.

**Model 1-"moral judgement".** In the first model "*moral judgement*" (MJ), agitated behaviors are assumed by staff to arise from the patient's own moral agency rather than an expression of any particular need or neurological symptoms. Thus, these are behaviors which are perceived as negative and directed towards staff who are obliged to endure them or avoid them. Paradoxically, agitation was seen as 'bad' and sometimes simultaneously, meaningless behavior. Staff who tended to adopt a 'judgement' of agitation behavior described it as unpredictable and frightening. Commonly, patients were described as "violent", "abusive", "aggressive", "threatening" and "trying to abscond". For example, during an observation in Hospital A (0102/01) one HCA described an agitated patient's behavior as purposeful and suggested that they could control it:

> "*This patient is being too violent, everything, we can't stop him from leaving, we can't stop him from leaving the ward, because some people, those who are mobile, they may try to abscond, but if they're not doing that and if the security comes out they calm down.*" [Hospital A, HCA 102]

In the MJ model we noted that staff focused almost exclusively on the patients' physical needs, but rarely explored possible sources of agitation when they occurred. While there were instances of compassionate care, such interactions seemed instrumental–that is, to prevent further agitation and on such occasions, patients were admonished for previous 'bad' behavior. Withdrawal and avoiding direct contact could be explained by staff fears of agitation as highly unpredictable and frightening.

> "*They become very violent towards the nurses as well so we have to be very careful as well*" [Hospital A, HCA 201]

At times, this avoidance tended to exacerbate the agitation. In one observation at hospital A (0102/3) a nurse entered to write medication refusal in [participant's] notes; stating that she hadn't worked with this gentleman before and that that she was scared "he will kick out". She says she doesn't blame them, "it's hard work, I don't think you should bother forcing people like that".

Similar attitudes were articulated at other sites. Unpredictability, limited training and knowledge on the causes of agitation appear to increase staff fears of being hit. For example, such fears were expressed by a care home music therapist who said that she had not received any formal training:

> "*Well as sort of being frightened by maybe the kind of unpredictability of it or the potential sort of unexpected violence or strength that sometimes people show. I mean it's just frightening because it's kind of, I don't know.*" [Care home C, music therapist 605)

## Model 2 –needs-based model

In contrast to a moral judgement model, in the "needs-based" (NB) model, there is a tendency to regard agitation as part of dementia as a disease, distressing to the patient, and potentially amenable to intervention. Staff therefore had a proactive, investigative approach seeking to detect possible causes and remedies. We noted that staff who adopt this model, approached people with agitation with a degree of compassionate detachment rather than avoidance or criticism.

In the needs-based model, a wide range of causes of agitation were suggested, [19]: Physical reasons (i.e. pain, thirst, uncomfortable, constipation), emotional causes (i.e. boredom,

frightened, disoriented, missing home), environmental challenges (i.e. unfamiliar surroundings, change of routine, noise and light), cognitive reasons (disorientation, difficulty in expressing needs) and social reasons (family visiting or not visiting, copying other agitated patients, wanting attention, and poor communication by staff).

> *Some of us understand that the person will not shout for just no reason. So, our business is to find out why. A person may not, I mean, attempt to hit you for absolutely no reason. It's your business to find out why. Or, the individual may not decide to be uncooperative with personal care for absolutely no reason. It's your business, really, to... Well, what I might... It's about reflecting on the job as you're doing it, so you're thinking, I'm trying to clean his teeth, for instance, and he's not agreeing; he's closing his mouth. What am I doing wrong? [Care home C, RMN 601]*

We also noted that this model appeared to promote a more empathic understanding of how the person with dementia may be feeling and also a consideration of their previous life context:

> *"If you change place with this person for five minutes, just, you take up all of his condition and everything, you will know that it's a very unpleasant place to be. So, you must realise when you are working with them, that they are in an unpleasant place." [Care home C, RMN 601]*

> *"So, our job as a nurse is to give them that assurance, that support, empathy, show them . . . give them empathy and help them in whatever way because it depends on each individual needs. Can be different most times so some of them with music it helps them to remember what . . . how they led their lives and it helps them to remember who they really are. At least at that moment in time while some of them is that they just want somebody to talk to, you understand." [Hospital B, SN 402]*

We noted distinct differences in the pervasiveness of MJ and NB models within the different settings and these appeared to strongly influence agitation responses, individually and institutionally. In hospital B and care homes C and D where agitation was framed in terms of being a response to an unmet need, staff used more comforting speech, gentle distraction, and what they knew about the person, which was often learned from care documentation. Staff at hospital B were typically more relaxed around patients than staff at hospital A, and more knowledgeable about them.

> *"We're not to use the word, challenging. So it's more about anxiety-behavior, anxiety management." [Site D, Care manager 702]*

In another observation at hospital B (0504/01) we noted: Ward sister approaches and brings sheet "can we cover you up? Is that comfortable?" I cannot hear answer. 'Do you want to sit in the chair?' [Participant] agrees "okay" and touches foot. Ward sister then puts some notes away. Participant plays with TV and ward sister looks around (as if concerned she is getting out of bed). Ward sister helps her get out of bed and puts second sock on, chatting to her. Female doctor arrives and talks to ward sister. Male doctor joins and immediately says hi and makes small talk with participant. They stay approximately 1 min, ward sister stays sitting on bed next to her chatting and gives her a knitted activity toy with ribbons and buttons to fiddle with ("twiddle-muff"). They talk about it.'

Increased staff knowledge of the patients reduced the potential for their objectification while enhancing confidence. In the care homes, where staff cared for the same individuals longer-term, there was even less fear. We also found that tactile comfort was offered more in care homes than in hospitals: Observation at care home C (0601/02): "Shouts 'stop it, stop it, stop it'. Very loudly. Air mattress in chair keeps moving and making a noise. Nurse approaches 01. Speaks very calmly and kindly. Uses touch, arm and cheek, gently to let him know he is there."

*". . .and sometimes they say to you, we have this training where they say you mustn't really hug them and whatever the case may be, sometimes the hugging does help. But it's for you to know who it is and the family member will enlighten you as well about the person, what the person was like before." [Care home D, HCA 703]*

Staff expressed positive regard for people with dementia and this extended to staff taking a position where they saw themselves as advocates for the person with dementia. This led to them defending the person with dementia and "speaking up" for them to other staff. This was expressed more frequently by care home staff.

*"And we're, you know, we're the. . . Some of them, we're the only voices they know, you know? So we really need to show them some conditional—what do you call it now?—unconditional positive regards for them." [Care home C, RMN 601]*

## Discussion and implications

This study provides unique insights into the care of agitation in people with dementia who are nearing the end of life and provides evidence on systemic factors that may exacerbate or attenuate these challenging phenomena. We found that the understanding of, and responses to, agitation behavior among care staff can vary significantly but these appear to be culturally embedded within different institutions. Using methods of triangulating structured observations, in-depth interviews and quantitative observations, we were able to identify models (patterns of attitudes and responses) that may be reasonably associated with different care settings. Moreover, we are able to identify structural-cultural issues within such settings that make negative responses to agitation much more likely. In doing so, it is not our intention to highlight the shortcomings of staff in such settings, but rather to suggest that these problems are systemic, most likely arising from the lack of value, priority and thus resources that organizations give to these areas. Thus, we found that a "moral judgement" model of agitation is more commonly observed within an organizational setting in which dementia care was not prioritized, staff received inadequate training and support and the physical environment was inappropriate for patients with dementia to the point of neglect. In the contrasting "needs-based" model in which agitation is regarded by care staff as an inherent aspect of the disease, the response is more likely to be swift, proactive and investigative.

Crucial to our understanding of the contextual issues surrounding sub-optimal care, staff felt stigmatized by working in older people's care, disempowered and frightened by people with agitation. In understaffed units, the sense of neglect of staff and patients appears pervasive. Within a "moral judgement" model, agitated behaviors are attributed to the patient's own moral agency (hence, there is an external focus on 'abusive' behavior to staff) and while this may occur randomly, staff feel victimized and then respond reproachfully or avoid contact. Being detached or avoidant, as ways of coping due to low confidence in managing one's role, are often associated with work-related stress and burnout [20, 21]. Furthermore, Pillemer and

Moore [22] found that high levels of patient to staff aggression and inadequate training on its management, increases the prevalence of abuse towards patients by staff. Paveza et al. [23], also found that aggression from patients with dementia increased their risk of being mistreated in community settings. The literature suggests that staff awareness of the expectations of the caring role may result in feelings of guilt or shame and where dissonance exists between expectations and actual behavior [9], some form of resolution may be attempted through denial of personhood in the patient; i.e. a belief that the patient doesn't have any feelings or true consciousness [24]. Such attempts tend to be temporary and ineffectual.

Positional theory as described by Harré et al. [25] refers to cognitive processes that are instrumental in supporting the actions that people undertake within local moral domains and which have the appearance of beliefs and practices involving rights and duties. Thus, particular norms and constraints within which practices occur, are determined "not by individual levels of competence alone, but by having rights and duties in relation to items in the local corpus of sayings and doings". Thus, how people think, behave, feel and perceive are only permissible or acceptable within specific contexts and these are usually 'taken for granted'. Thus, in positional theory, language is the primary platform and instrument of thought and social action, and it is through language that 'taken for granted' worlds are developed and established [26]. Typical staff discourses and language within the different wards revealed but also created quite distinct sets of beliefs and values about older people, dementia, agitation and response. Thus, in the problematic settings, staff were primed for distrust and anxiety and this was reflected in their approach to people with agitation where chastising or ignoring the person were more common responses.

Stevens et al [27] suggest that positional theory may help explain mistreatment in care settings and although we make no claims for mistreatment in the current study, it is reasonable nevertheless to consider avoidance as neglect, and certainly a negative behavior. Kitwood, Harré and Moghaddam [28] use the term 'malevolent positioning' to describe how people with dementia can be denied agency by others, usually by reference to their limited cognitive capacity. Various commentators [27, 29] highlight the disengagement from people with dementia adopted by care workers and others, in consequence of characterizations and perceptions that such people are not capable of being engaged. Moreover, positioning of this type becomes entrenched in settings where (according to the narrative in these contexts) staff shortages, and the remaining demotivated and poorly trained staff are discouraged from wasting time and effort on people who can't respond. This may be compounded by the low levels of trust and high levels of insecurity that we found among staff in a particular setting, exemplified by the casual and unconcerned language of patient aggression and violence used by these staff. That such staff are temporary, commonly low-paid and from migrant communities and whose first language is not English, may add to their sense of distance from the patients. Our findings suggest that staff in these settings feel similarly poorly nurtured by the employing organization and arguably, denied their own personhood [11].

## Strengths and limitations

Skilled and experienced clinical researchers made detailed observations at varying times of day and during the night, with coding of activity and interactions every two minutes. Our robust approach allowed triangulation of structured observations, field-notes and interviews, providing a deep connection between systems, events, beliefs and behaviors. We were, however, able to conduct an in-depth examination in only a small number of acute hospital and wards and cares homes. While a Hawthorn effect may have occurred, in which staff, consciously or unconsciously behaved differently due to the presence of the researchers, we spent time

acclimatizing to settings allowing staff to revert to usual behavior. Observations usually lasted for more than an hour and it would be challenging for staff to maintain change in practice over this time [30]. Staff who volunteered to participate in qualitative interviews may have been more aware and willing to discuss these issues and provided more socially desirable responses during the interviews. For example, staff did not mention the use of antipsychotic and sedative medications in managing people with agitation. This may be because they understand, in theory, these are to be avoided where possible, when in reality, and outside of a one to one interview staff may use these drugs in their practice. We did not interview any senior managerial staff to understand organizational perspectives which may have shed light on financial constraints, staffing and retention issues and wider political factors.

## Implications

Our findings highlight the complexity of caring for people with dementia who are agitated. Individual members of staff and units of care, i.e. wards and care homes, used explanatory models that spanned those we describe. Some individual members of staff could hold both "moral judgment" and "person-centered" explanatory models simultaneously. This may reflect the people with dementia they care for, who need calm and order, but behave in a way that is regarded as disruptive [31, 32].

We suggest that holding and managing these explanatory models simultaneously may lead to cognitive dissonance in staff, which increases burnout and emotional exhaustion and leads to them paying less regard to person-centered care [9, 32, 33], manifested in our observations as "looking and not intervening" and "chastising speech".

Organizational support and leadership are key drivers of good quality dementia and end of life care in care homes and acute hospitals [9, 34]. We found this to be an overarching factor in how staff managed agitation. Other structural factors such as staffing levels and environment and how staff who work with people with dementia are valued are vital [34]. This reflects Kitwood's view that staff must be cared for themselves to be able to look after people with dementia [24]. Many staff reported feeling stigmatized and intimidated by managers–key aspects of Kitwood's construct of malignant social psychology. This produces further malignant interactions when people are agitated; this can lead to increased staff disengagement, chastising speech and further reduces personhood.

The practice implications of our study centre on how we can "shift" staff between the two explanatory models, from "moral judgement" to "needs-based". Many studies have highlighted how training alone in both hospitals and care homes is not enough to produce maintained improvements in care [31, 35]; joint working with professionals experienced in dementia care, clinical role modelling, and supervision are vital [34, 36].

Managing agitation in a more person-centered way will improve both quality of life and of dying. Organizational ethos and support will be key to this change. Staff in care homes feel constrained by care home structures, processes and a culture of fear and scrutiny [9]; hospitals are driven by organizational targets that prioritises processes and patient "throughput" [32]. System changes must take into account the needs of staff as well as the person with dementia.

## Appendix 1: Qualitative interview topic guide

### Topic guide

"Thank you for agreeing to talk with me. This part of our study is to help us understand better how to care for people with dementia. Quite commonly, people with dementia experience a considerable level of agitation–they seem to be distressed but it is not very clear what, if

anything, is causing this to happen. You may have observed these episodes and we would really appreciate hearing your perceptions and views."

"In order to make sure that I don't miss anything, I will record our conversation on a digital recorder and then it will be professionally transcribed. Once the interview has been transcribed, I will ensure that all identifying characteristics are removed. Once again, everything you tell me will be treated with complete confidence."

Interview

Before we begin, just for the record, could you tell me a little about yourself (your relationship with the patient/resident; if you have been the main carer, how long you have been caring, do you have help from other people–that sort of thing. . ..) If you are a member of staff, where have you worked I the past and what is your experience.

Probe for employment and social support

## Explanatory models

When did you first notice your (relationship) becoming agitated?

Tell me about what happens when your (relationship) becomes agitated?

How would you describe these occasions?

Probe for: Different types of agitation and any patterns–e.g. do they occur at particular times?

What do think causes these episodes?

Probe: Do you think that different things may cause them; what might these be? Suggest with pain or during care tasks

How long do these episodes last?

Is there anything that you feel helps to reduce or resolve the agitation?

Probe for all the things that the participant does–if anything (they may feel it best to leave the person alone but don't make it appear that this is wrong)

Impact on carer (ignore, if the participant has already described the impact of these episodes)

How do these episodes make you feel?

Probe for emotional impact–(N.B. however, researcher should not assume that the participant feels distressed)

How do you cope with these episodes? Are there things that you can do to help you feel better?

Have you talked about these episodes with other people?

What do they think causes them?

## Appendix 2: Checklist of agitated behaviors: Cohen-Mansfield Agitation Inventory

### Agitation type

1. Pace, aimless wandering

2. Inappropriate dress or disrobing

3. Spitting (include at meals)

4. Cursing or verbal aggression

5. Constant unwarranted request for attention or help

6. Repetitive sentence or questions

7. Hitting (include self)

8. Kicking

9. Grabbing onto people

10. Pushing

11. Throwing things

12. Strange noises (weird laughter or crying)

13. Screaming

14. Biting

15. Scratching

16. Trying to get to a different place (e.g. out of the room, building)

17. Intentional falling

18. Complaining

19. Negativism

20. Eating/drinking inappropriate substances

21. Hurt self or other (cigarette, hot water etc.)

22. Handling things inappropriately

23. Hiding things

24. Hoarding things

25. Tearing things or destroying property

26. Performing repetitious mannerisms

27. Making verbal sexual advances

28. Making physical sexual advances

29. General restlessness

## Acknowledgments

We wish to thank all the care home staff who participated in this study and all the Alzheimer's Society research network volunteers, whose lives have been affected by dementia, for their contribution.

## Author Contributions

**Conceptualization:** Elizabeth L. Sampson, Aisling Stringer, Francesca La Frenais, Gill Livingston.

**Formal analysis:** Elizabeth L. Sampson, Aisling Stringer, Francesca La Frenais.

**Funding acquisition:** Elizabeth L. Sampson, Gill Livingston, Gerard Leavey.

**Investigation:** Francesca La Frenais, Shanlee Higgins, Mary-Jo Doyle, Anne Laybourne.

**Methodology:** Gerard Leavey.

**Supervision:** Elizabeth L. Sampson, Gill Livingston, Gerard Leavey.

**Writing – original draft:** Elizabeth L. Sampson, Aisling Stringer, Francesca La Frenais, Shanlee Higgins, Mary-Jo Doyle, Anne Laybourne, Gill Livingston, Gerard Leavey.

**Writing – review & editing:** Elizabeth L. Sampson, Aisling Stringer, Francesca La Frenais, Shanlee Higgins, Mary-Jo Doyle, Anne Laybourne, Gill Livingston, Gerard Leavey.

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
