## [Decision Letter · Decision Letter 0]

30 Jul 2019

PONE-D-19-18618

Agitation near the end of life with dementia: an ethnographic and structured observational study of care

PLOS ONE

Dear Professor Sampson,

Thank you for submitting your manuscript to PLOS ONE. After careful consideration, we feel that it has merit but does not fully meet PLOS ONE’s publication criteria as it currently stands. Therefore, we invite you to submit a revised version of the manuscript that addresses the points raised during the review process.

We would appreciate receiving your revised manuscript by Sep 13 2019 11:59PM. To enhance the reproducibility of your results, we recommend that if applicable you deposit your laboratory protocols in protocols.io, where a protocol can be assigned its own identifier (DOI) such that it can be cited independently in the future. For instructions see: http://journals.plos.org/plosone/s/submission-guidelines#loc-laboratory-protocols

We look forward to receiving your revised manuscript.

Kind regards,

Tim Luckett

Academic Editor

PLOS ONE

4. We note you have included a table to which you do not refer in the text of your manuscript. Please ensure that you refer to Table 3 in your text; if accepted, production will need this reference to link the reader to the Table.

Reviewers' comments:

Reviewer's Responses to Questions

**Comments to the Author**

1. Is the manuscript technically sound, and do the data support the conclusions?

Reviewer #1: Partly

Reviewer #2: Yes

2. Has the statistical analysis been performed appropriately and rigorously? 

Reviewer #1: N/A

Reviewer #2: N/A

3. Have the authors made all data underlying the findings in their manuscript fully available?

Reviewer #1: Yes

Reviewer #2: Yes

4. Is the manuscript presented in an intelligible fashion and written in standard English?

Reviewer #1: Yes

Reviewer #2: Yes

5. Review Comments to the Author

Reviewer #1: Thank you for the opportunity to review your manuscript describing a multi-method study of health carers' response to agitation in people with moderate to severe dementia.

The focus of your research is important, and the methods at first glance seemed appropriate to bring the often hidden care of patients into the light of day for critical examination. The study has potential to improve understanding of what constitutes high quality responsive care of people with agitated behaviours in dementia, what contributes and detracts from quality and why there is variation. It is also clear that a significant amount of work has been undertaken in the study.

However, there are several areas requiring attention before the manuscript is ready for publication. The main issues to be addressed are:

1. Greater detail of reporting of the methods, including analysis and data integration is needed. For example, the study is referred to as ethnographic in the title, but there is no definition or even mention of the ethnographic method in the body of the manuscript. The authors have outlined the three different data collection methods well, but again, these are discrepant from what is referred to in the title. There is an assertion that innovative and robust triangulation of the data was undertaken (Lines 522 and 586), but there is no description of the data integration/triangulation process in the methods section, nor could I grasp it from the way the results are currently reported. The analysis section requires more detail of exactly how the three different data sets were analysed and then integrated/triangulated.

2. The results are reported in the wrong order. It would make more sense for the reader (and be more conventional) to report results about the type of staff, then types of behaviours observed, then key themes, then the explanatory model. I suggest that the explanatory model is the result arising from triangulation/integration of the data, hence why it makes more sense for it to be reported as your final interpretation of the multiple results. Also, the key themes should be named to make it clearer to the reader what they actually were.

3. There are a number of different terms, concepts and models referred to throughout the manuscript that were not clearly defined and some I suggest unnecessarily complicate the writing and result in a confusing rather than clarifying effect. For example, 'ideal types' is a sociological term borrowed from Weber which has a different meaning to the lay understanding of 'ideal'. As Model 1 clearly is not an ideal way to care for people (according to lay understanding of the word ideal), what is the advantage in using this sociological term? I found the section on 'positional theory' (lines 557-570) to be confusing and nor did I understand its connection to the models you have proposed. Other key terms/concepts that should be better defined, explained or substituted with simpler terms are 'end of life', 'personhood', 'symbolic interactionist', 'materialistic stance' and 'moral agency'.

Other points to address:

4. There was discrepancies in how the observed people with dementia are referred to: at times they are called 'non-participants' and other times they are called 'participants'. Please revise these sections for consistency.

5. The abstract does not explicitly name the first proposed model, only the second.

6. Lines 70 - 70 contain some overly long and confusing sentences - suggest revise for improved clarity of meaning.

7. More detail is needed on the Noticeable Problems Checklist.

8. It's not clear why you chose to focus only on people with advanced dementia in your study. There is a brief mention of this line 68-9 (and again at 626), but none of the results explicitly refer to scenarios of dying. Greater justification and explanation of the relevance of your study to end of life care is needed.

9. Lines 136-139: were ethical concerns also discussed with site managers? If so, at what point of the study? If not, why not?

10. Table 1 and 2 - staff and patient demographics should be summarised so as not to identify individuals. E.g. x % male, etc.

11. Given that pharmacological intervention for agitated behaviours in dementia are well documented as common clinical practice (although not necessarily best practice), why was there no mention of this type of intervention by staff in your results?

12. Line 546: Abuse to whom and by whom?

13. Lines 548-552: I don't believe that this assertion is clearly evidenced by the data presented.

14. Line 622: The term 'moral judgement' differs from the name of model 1, which is 'moral assessment'. Please revise for consistency.

15. There are a few typographical errors and missing words throughout, including in Appendix 1, which I am sure the authors will identify and correct as part of their revision.

I wish you well in the revision of this study report and your ongoing work in this important aspect of caring for people with dementia.

Reviewer #2: Thank you for the opportunity to review this paper. Agitation is a common and challenging issue in care home settings and this study offers a valuable insight into how staff perceive agitation, which will support development and implementation of interventions that may help staff to recognise, prevent and support agitated behaviours. I have a few areas where further clarity would help the reader.

1. On lines 93-7 the authors state that they appointed a personal consultee in order to gain agreement for the participants with dementia to take part in the study. This was because they had been assessed to lack capacity to give informed consent (which would likely be indicated by the severity of dementia of those being recruited), but I wonder if this should be made clear for readers not familiar with the MCA? Likewise a consultee provides advice on what the person’s wishes would have been if they had capacity. The use of the word agreement might suggest some kind of proxy consent, so again just for the purposes of clarity around capacity and consent processes in England and Wales for the wider readership of this journal, I wonder if it would be helpful to also clarify this? Also would it be useful to state the correct technical name for the professional consultee i.e. nominated consultee, given this is provided for personal consultees?

2. It would be useful to provide details of how the researchers decided when to observe participants in terms of the time periods they spent within each care setting and over what period of time the observations took place if there was more than one for a participant.

3. In table 3 it would be useful to provide the units for the final column, I’m presuming this is total minutes over the observation periods but this is not clear.

4. On line 249 the authors state patients had a ‘forget-me-not’ magnet on their board. It may be helpful to explain this further for the international readership of this journal who may be unfamiliar with the flower and its meaning in a UK context.

5. It was interesting that in Hospital A, staff predominantly took a MA response to agitated behaviours which was largely a passive rather than active approach, yet the % of the observed period where patients were agitated was the lowest across the four sites. Likewise in site C where staff were most likely to take active and comforting approaches to agitation the percentage of time residents were agitated was the highest. Why might this be? I feel this warrants some further discussion in the context of the participants, models, approaches and institutional contexts within the paper.

6. The sentence on lines 349-351 needs reviewing.

7. Lines 553-4 there is a ) missing here.

8. Lines 560-1 the page number for the quotation is needed

9. I would find it helpful if the authors could say more about how positional theory might relate to how moral landscapes of what is seen as permissible or not prevail within organisations and thus how this might be addressed in the case of settings where predominantly ‘chastising’ or ‘looking but not intervening’ approaches are prevalent. There is one sentence that addresses this, but I feel the section would benefit more from discussion of this.

10. Line 603 uses different terms of the models to those given earlier in the paper. It might be helpful to stick to the same terms throughout?

11. The para lines 621-626 discusses the need to try and move people to a more needs-based model of agitated behaviour assessment, however, given the context that higher levels of agitation were displayed in the settings where staff provided more comforting responses, there is a potential implication that by doing so staff may not in fact reduce agitation levels and thus the burnout risk could potentially be higher. Thus on further implication might be that any movement of staff perceptions will need to be accompanied by interventions that can potentially equip staff with a greater range of skills and resources to help prevent and reduce agitation.

6. PLOS authors have the option to publish the peer review history of their article (what does this mean?). If published, this will include your full peer review and any attached files.

Reviewer #1: No

Reviewer #2: Yes: Professor Claire A Surr

---

## [Author Response · Author response to Decision Letter 0]

9 Aug 2019

Please see the detailed document "Response to reviewers comments"

---

## [Decision Letter · Decision Letter 1]

6 Sep 2019

[EXSCINDED]

PONE-D-19-18618R1

Agitation near end of life with dementia: an ethnographic study of care

PLOS ONE

Dear Professor Sampson,

Thank you for submitting your manuscript to PLOS ONE. After careful consideration, we feel that it has merit but does not fully meet PLOS ONE’s publication criteria as it currently stands. Therefore, we invite you to submit a revised version of the manuscript that addresses the points raised during the review process.

I am satisfied that you have addressed comments from Reviewer 2. However, Reviewer 1 has some further suggestions for improving the quality of your manuscript. I especially agree with this reviewer that more details should be provided for the analysis, in a similar way that you have done for synthesis following the last round of reviews.

We would appreciate receiving your revised manuscript by Oct 21 2019 11:59PM. To enhance the reproducibility of your results, we recommend that if applicable you deposit your laboratory protocols in protocols.io, where a protocol can be assigned its own identifier (DOI) such that it can be cited independently in the future. For instructions see: http://journals.plos.org/plosone/s/submission-guidelines#loc-laboratory-protocols

We look forward to receiving your revised manuscript.

Kind regards,

Tim Luckett

Academic Editor

PLOS ONE

Reviewers' comments:

**Comments to the Author**

1. If the authors have adequately addressed your comments raised in a previous round of review and you feel that this manuscript is now acceptable for publication, you may indicate that here to bypass the “Comments to the Author” section, enter your conflict of interest statement in the “Confidential to Editor” section, and submit your "Accept" recommendation.

Reviewer #1: (No Response)

2. Is the manuscript technically sound, and do the data support the conclusions?

Reviewer #1: Partly

3. Has the statistical analysis been performed appropriately and rigorously? 

Reviewer #1: No

4. Have the authors made all data underlying the findings in their manuscript fully available?

Reviewer #1: Yes

5. Is the manuscript presented in an intelligible fashion and written in standard English?

Reviewer #1: Yes

6. Review Comments to the Author

Reviewer #1: Thank you for the opportunity to review your revised manuscript. The revisions have resulted in an improved and clearer version, particularly with respect to the discussion section, and the topic is one of clinical and ethical importance. I also appreciate that the authors have highlighted the need to respect and address the needs of staff caring for people with advanced dementia in order to improve the compassion and effectiveness of their care of people with agitated behaviours. For these reasons, I hope this work will be published at some point. However, I was disappointed to see that not all of the reviewers' comments were adequately addressed and there remains major problems with the reporting of data analysis and results, along with several more minor issues that in combination demonstrate a lack of attention to detail and process.

The major issues are as follows:

1. Qualitative and quantitative data analysis and integration: The reporting of analysis methods is insufficient. For example, Excel, NVivo and STAT are tools to help manage and store data, not data analysis per se. Please report the exact analysis methods. Line 239 state you used graphs, networks and diagrams to synthesis the data, but none of these are present in the report. In addition, the authors response to this item does not contain the correct line numbers for the sections referred to.

2. Findings/results: Examples of poor reporting: Table 1, Site B, Role column has '1 senior nurse' listed twice. Table 2 data requires further summarisation e.g mean age, total number of observations per site, total time spent agitated per site.Table 3 data simply does not make sense, as % do not add up to 100% either horizontally or vertically. There is an error (% written twice) in the header of column 4. The 'Response to agitation' theme was positioned as reporting staff interview data and yet reports observational data as evidence, which in addition does not identify who the person is whose words are quoted. Overall, as the reporting of the findings is confusing and lacks attention to detail and method, I lost faith in the veracity of the processes of data analysis.

More minor issues:

1. The term 'non-participant observations' continues to be confusing as it is often used in the same sentence as 'participants'. While it may be a common term in ethnographic research, I suggest that 'structured observation' would be sufficient and also cause less cognitive dissonance from the point of view of the reader.

2. Lines 80-81: what are the exact prevalence of agitated behaviours in people with more advanced dementia (i.e. needs more specific detail that simply 'common').

3. My previous point about sharing ethical concerns with managers referred to concerns that arose during the study, not potential ones. What was the process for reporting back to managers issues of concern witnessed by the researchers?

4. Objectives are stated for the observation aspect of the study, but not for the interviews.

5. Many punctuation, spelling, grammatical and typographical errors remain. For example, there is use of both American and UK English spelling, use of '&' rather than 'and', multiple spacings between words, and missing or misplaced commas.

Taken together, these issues greatly detract from the quality of the study report and require addressing before the manuscript is worthy of publication. I wish you well in your ongoing revisions in order that your work ultiately leads to improved care for people with advanced dementia.

7. PLOS authors have the option to publish the peer review history of their article (what does this mean?). If published, this will include your full peer review and any attached files.

Reviewer #1: No

---

## [Author Response · Author response to Decision Letter 1]

2 Oct 2019

Please see the letter to the editor and the response to reviewer documents.

---

## [Editor Report · Decision Letter 2]

4 Oct 2019

Agitation near end of life with dementia: an ethnographic study of care

PONE-D-19-18618R2

Dear Dr. Sampson,

We are pleased to inform you that your manuscript has been judged scientifically suitable for publication and will be formally accepted for publication once it complies with all outstanding technical requirements.

With kind regards,

Tim Luckett

Academic Editor

PLOS ONE

---

## [Editor Report · Acceptance letter]

11 Oct 2019

PONE-D-19-18618R2 

Agitation near the end of life with dementia: An ethnographic study of care 

Dear Dr. Sampson:

I am pleased to inform you that your manuscript has been deemed suitable for publication in PLOS ONE. Congratulations! Your manuscript is now with our production department. 

With kind regards,

on behalf of

Dr. Tim Luckett 

Academic Editor

PLOS ONE